# PAIRNORM: TACKLING OVERSMOOTHING IN GNNS

**Lingxiao Zhao**
Carnegie Mellon University
Pittsburgh, PA 15213, USA
{lingxia1}@andrew.cmu.edu

**Leman Akoglu**
Carnegie Mellon University
Pittsburgh, PA 15213, USA
{lakoglu}@andrew.cmu.edu

## ABSTRACT

The performance of graph neural nets (GNNs) is known to gradually decrease with increasing number of layers. This decay is partly attributed to oversmoothing, where repeated graph convolutions eventually make node embeddings indistinguishable. We take a closer look at two different interpretations, aiming to quantify oversmoothing. Our main contribution is PAIRNORM, a novel normalization layer that is based on a careful analysis of the graph convolution operator, which prevents all node embeddings from becoming too similar. What is more, PAIRNORM is fast, easy to implement without any change to network architecture nor any additional parameters, and is broadly applicable to any GNN. Experiments on real-world graphs demonstrate that PAIRNORM makes deeper GCN, GAT, and SGC models more robust against oversmoothing, and significantly boosts performance for a new problem setting that benefits from deeper GNNs. Code is available at https://github.com/LingxiaoShawn/PairNorm.

## 1 INTRODUCTION

Graph neural networks (GNNs) is a family of neural networks that can learn from graph structured data. Starting with the success of GCN (Kipf & Welling, 2017) on achieving state-of-the-art performance on semi-supervised classification, several variants of GNNs have been developed for this task; including GraphSAGE (Hamilton et al., 2017), GAT (Velickovic et al., 2018), SGC (Wu et al., 2019), and GMNN (Qu et al., 2019) to name a few most recent ones.

A key issue with GNNs is their depth limitations. It has been observed that deeply stacking the layers often results in significant drops in performance for GNNs, such as GCN and GAT, even beyond just a few (2–4) layers. This drop is associated with a number of factors; including the vanishing gradients in back-propagation, overfitting due to the increasing number of parameters, as well as the phenomenon called oversmoothing. Li et al. (2018) was the first to call attention to the oversmoothing problem. Having shown that the graph convolution is a type of *Laplacian smoothing*, they proved that after repeatedly applying Laplacian smoothing many times, the features of the nodes in the (connected) graph would converge to similar values—the issue coined as "*oversmoothing*". In effect, oversmoothing hurts classification performance by causing the node representations to be indistinguishable across different classes. Later, several others have alluded to the same problem (Xu et al., 2018; Klicpera et al., 2019; Rong et al., 2019; Li et al., 2019) (See §5 Related Work).

In this work, we address the oversmoothing problem in deep GNNs. Specifically, we propose (to the best of our knowledge) *the first normalization layer for GNNs* that is applied in-between intermediate layers during training. Our normalization has the effect of preventing the output features of distant nodes to be too similar or indistinguishable, while at the same time allowing those of connected nodes in the same cluster become more similar. We summarize our main contributions as follows.

- **Normalization to Tackle Oversmoothing in GNNs:** We introduce a normalization scheme, called PAIRNORM, that makes GNNs significantly more robust to oversmoothing and as a result enables the training of deeper models without sacrificing performance. Our proposed scheme capitalizes on the understanding that most GNNs perform a special form of Laplacian smoothing, which makes node features more similar to one another. The key idea is to ensure that the total pairwise feature distances remains a *constant* across layers, which in turn leads to distant pairs having less similar features, preventing feature mixing across clusters.

- **Speed and Generality:** PAIRNORM is very straightforward to implement and introduces no additional parameters. It is simply applied to the output features of each layer (except the last one) consisting of simple operations, in particular centering and scaling, that are *linear* in the input size. Being a simple normalization step between layers, PAIRNORM is not specific to any particular GNN but rather applies broadly.
- **Use Case for Deeper GNNs:** While PAIRNORM prevents performance from dropping significantly with increasing number of layers, it does not necessarily yield increased performance in absolute terms. We find that this is because shallow architectures with no more than 2–4 layers is sufficient for the often-used benchmark datasets in the literature. In response, we motivate a real-world scenario wherein a notable portion of the nodes have *no* feature vectors. In such settings, nodes benefit from a larger range (i.e., neighborhood, hence a deeper GNN) to "recover" effective feature representations. Through extensive experiments, we show that GNNs employing our PAIRNORM significantly outperform the 'vanilla' GNNs when deeper models are beneficial to the classification task.

## 2 UNDERSTANDING OVERSMOOTHING

In this work, we consider the semi-supervised node classification (SSNC) problem on a graph. In the general setting, a graph $\mathcal{G} = (\mathcal{V}, \mathcal{E}, \mathbf{X})$ is given in which each node $i \in \mathcal{V}$ is associated with a feature vector $\mathbf{x}_i \in \mathbb{R}^d$ where $\mathbf{X} = [\mathbf{x}_1, \ldots, \mathbf{x}_n]^T$ denotes the feature matrix, and a subset $\mathcal{V}_l \subset \mathcal{V}$ of the nodes are labeled, i.e. $y_i \in \{1, \ldots, c\}$ for each $i \in \mathcal{V}_l$ where $c$ is the number of classes. Let $\mathbf{A} \in \mathbb{R}^{n \times n}$ be the adjacency matrix and $\mathbf{D} = \text{diag}(deg_1, \ldots, deg_n) \in \mathbb{R}^{n \times n}$ be the degree matrix of $\mathcal{G}$. Let $\tilde{\mathbf{A}} = \mathbf{A} + \mathbf{I}$ and $\tilde{\mathbf{D}} = \mathbf{D} + \mathbf{I}$ denote the augmented adjacency and degree matrices with added self-loops on all nodes, respectively. Let $\tilde{\mathbf{A}}_{\text{sym}} = \tilde{\mathbf{D}}^{-1/2} \tilde{\mathbf{A}} \tilde{\mathbf{D}}^{-1/2}$ and $\tilde{\mathbf{A}}_{\text{rw}} = \tilde{\mathbf{D}}^{-1} \tilde{\mathbf{A}}$ denote symmetrically and nonsymmetrically normalized adjacency matrices with self-loops.

The task is to learn a hypothesis that predicts $y_i$ from $\mathbf{x}_i$ that generalizes to the unlabeled nodes $\mathcal{V}_u = \mathcal{V} \backslash \mathcal{V}_l$. In Section 3.2, we introduce a variant of this setting where only a subset $\mathcal{F} \subset \mathcal{V}$ of the nodes have feature vectors and the rest are missing.

### 2.1 THE OVERSMOOTHING PROBLEM

Although GNNs like GCN and GAT achieve state-of-the-art results in a variety of graph-based tasks, these models are not very well-understood, especially why they work for the SSNC problem where only a small amount of training data is available. The success appears to be limited to shallow GNNs, where the performance gradually decreases with the increasing number of layers. This decrease is often attributed to three contributing factors: (1) overfitting due to increasing number of parameters, (2) difficulty of training due to vanishing gradients, and (3) oversmoothing due to many graph convolutions.

Among these, perhaps the least understood one is oversmoothing, which indeed lacks a formal definition. In their analysis of GCN's working mechanism, Li et al. (2018) showed that the graph convolution of GCN is a special form of Laplacian smoothing. The standard form being $(\mathbf{I} - \gamma \mathbf{I})\mathbf{X} + \gamma \tilde{\mathbf{A}}_{\text{rw}} \mathbf{X}$, the graph convolution lets $\gamma = 1$ and uses the symmetrically normalized Laplacian to obtain $\tilde{\mathbf{X}} = \tilde{\mathbf{A}}_{\text{sym}} \mathbf{X}$, where the new features $\tilde{\mathbf{x}}$ of a node is the weighted average of its own and its neighbors' features. This smoothing allows the node representations within the same cluster become more similar, and in turn helps improve SSNC performance under the cluster assumption (Chapelle et al., 2006). However when GCN goes deep, the performance can suffer from oversmoothing where node representations from different clusters become mixed up. Let us refer to this issue of node representations becoming too similar as *node-wise* oversmoothing.

Another way of thinking about oversmoothing is as follows. Repeatedly applying Laplacian smoothing too many times would drive node features to a stationary point, washing away all the information from these features. Let $\mathbf{x}_{\cdot j} \in \mathbb{R}^n$ denote the $j$-th column of $\mathbf{X}$. Then, for *any* $\mathbf{x}_{\cdot j} \in \mathbb{R}^n$:

$$\lim_{k \to \infty} \tilde{\mathbf{A}}_{\text{sym}}^k \mathbf{x}_{\cdot j} = \boldsymbol{\pi}_j \quad \text{and} \quad \frac{\boldsymbol{\pi}_j}{\|\boldsymbol{\pi}_j\|_1} = \boldsymbol{\pi} \, , \tag{1}$$

where the normalized solution $\boldsymbol{\pi} \in \mathbb{R}^n$ satisfies $\boldsymbol{\pi}_i = \frac{\sqrt{deg_i}}{\sum_i \sqrt{deg_i}}$ for all $i \in [n]$. Notice that $\boldsymbol{\pi}$ is independent of the values $\mathbf{x}_{\cdot j}$ of the input feature and is only a function of the graph structure (i.e.,

degree). In other words, (Laplacian) oversmoothing washes away the signal from *all* the features, making them indistinguishable. We will refer to this viewpoint as *feature-wise* oversmoothing.

To this end we propose two measures, row-diff and col-diff, to quantify these two types of oversmoothing. Let $\mathbf{H}^{(k)} \in \mathbb{R}^{n \times d}$ be the representation matrix after $k$ graph convolutions, i.e. $\mathbf{H}^{(k)} = \tilde{\mathbf{A}}_{\text{sym}}^k \mathbf{X}$. Let $\mathbf{h}_i^{(k)} \in \mathbb{R}^d$ be the $i$-th row of $\mathbf{H}^{(k)}$ and $\mathbf{h}_{\cdot i}^{(k)} \in \mathbb{R}^n$ be the $i$-th column of $\mathbf{H}^{(k)}$. Then we define row-diff($\mathbf{H}^{(k)}$) and col-diff($\mathbf{H}^{(k)}$) as follows.

$$\text{row-diff}(\mathbf{H}^{(k)}) = \frac{1}{n^2} \sum_{i,j \in [n]} \left\| \mathbf{h}_i^{(k)} - \mathbf{h}_j^{(k)} \right\|_2 \tag{2}$$

$$\text{col-diff}(\mathbf{H}^{(k)}) = \frac{1}{d^2} \sum_{i,j \in [d]} \left\| \mathbf{h}_{\cdot i}^{(k)} / \|\mathbf{h}_{\cdot i}^{(k)}\|_1 - \mathbf{h}_{\cdot j}^{(k)} / \|\mathbf{h}_{\cdot j}^{(k)}\|_1 \right\|_2 \tag{3}$$

The row-diff measure is the average of all pairwise distances between the node features (i.e., rows of the representation matrix) and quantifies node-wise oversmoothing, whereas col-diff is the average of pairwise distances between ($L_1$-normalized[1]) columns of the representation matrix and quantifies feature-wise oversmoothing.

## 2.2 STUDYING OVERSMOOTHING WITH SGC

Although oversmoothing can be a cause of performance drop with increasing number of layers in GCN, adding more layers also leads to more parameters (due to learned linear projections $\mathbf{W}^{(k)}$ at each layer $k$) which magnify the potential of overfitting. Furthermore, deeper models also make the training harder as backpropagation suffers from vanishing gradients.

In order to decouple the effect of oversmoothing from these other two factors, we study the oversmoothing problem using the SGC model (Wu et al., 2019). (Results on other GNNs are presented in §4.) SGC is simplified from GCN by removing all projection parameters of graph convolution layers and all nonlinear activations between layers. The estimation of SGC is simply written as:

$$\widehat{\boldsymbol{Y}} = \text{softmax}(\tilde{\mathbf{A}}_{\text{sym}}^K \mathbf{X} \mathbf{W}) \tag{4}$$

where $K$ is the number of graph convolutions, and $\mathbf{W} \in \mathbb{R}^{d \times c}$ denote the learnable parameters of a logistic regression classifier.

Note that SGC has a *fixed* number of parameters that does not depend on the number of graph convolutions (i.e. layers). In effect, it is guarded against the influence of overfitting and vanishing gradient problem with more layers. This leaves us only with oversmoothing as a possible cause of performance degradation with increasing $K$. Interestingly, the simplicity of SGC does not seem to be a sacrifice; it has been observed that it achieves similar or better accuracy in various relational classification tasks (Wu et al., 2019).

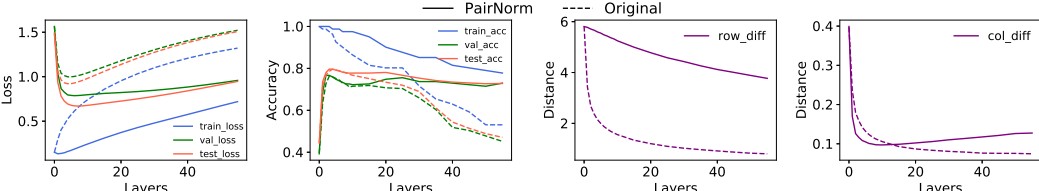

Figure 1: (best in color) SGC's performance (dashed lines) with increasing graph convolutions ($K$) on `Cora` dataset (train/val/test split is 3%/10%/87%). For each $K$, we train SGC in 500 epochs, save the model with the best validation accuracy, and report all measures based on the saved model. Measures row-diff and col-diff are computed based on the final layer representation of the saved model. (Solid lines depict after applying our method PAIRNORM, which we discuss in §3.2.)

Dashed lines in Figure 1 illustrate the performance of SGC on the `Cora` dataset as we increase the number of layers ($K$). The training (cross-entropy) loss monotonically increases with larger $K$, potentially because graph convolution mixes node representations with their neighbors' and makes them less distinguishable (training becomes harder). On the other hand, graph convolutions (i.e., smoothing) improve generalization ability, reducing the gap between training and validation/test loss

---

[1]We normalize each column $j$ as the Laplacian smoothing stationary point $\boldsymbol{\pi}_j$ is not scale-free. See Eq. (1).

up to $K = 4$, after which (over)smoothing begins to hurt performance. The row-diff and col-diff both continue decreasing monotonically with $K$, providing supporting evidence for oversmoothing.

# 3 TACKLING OVERSMOOTHING

## 3.1 PROPOSED PAIRNORM

We start by establishing a connection between graph convolution and an optimization problem, that is graph-regularized least squares (GRLS), as shown by NT & Maehara (2019). Let $\bar{\mathbf{X}} \in \mathbb{R}^{n \times d}$ be a new node representation matrix, with $\bar{\mathbf{x}}_i \in \mathbb{R}^d$ depicting the $i$-th row of $\bar{\mathbf{X}}$. Then the GRLS problem is given as

$$\min_{\bar{\mathbf{X}}} \sum_{i \in \mathcal{V}} \|\bar{\mathbf{x}}_i - \mathbf{x}_i\|_{\tilde{\mathbf{D}}}^2 + \sum_{(i,j) \in \mathcal{E}} \|\bar{\mathbf{x}}_i - \bar{\mathbf{x}}_j\|_2^2 \tag{5}$$

where $\|\mathbf{z}_i\|_{\tilde{\mathbf{D}}}^2 = \mathbf{z}_i^T \tilde{\mathbf{D}} \mathbf{z}_i$. The first term can be seen as total degree-weighted least squares. The second is a graph-regularization term that measures the *variation* of the new features over the graph structure. The goal of the optimization problem can be stated as estimating new "denoised" features $\bar{\mathbf{x}}_i$'s that are not too far off of the input features $\mathbf{x}_i$'s and are *smooth* over the graph structure.

The GRLS problem has a closed form solution $\bar{\mathbf{X}} = (2\mathbf{I} - \tilde{\mathbf{A}}_{\mathrm{rw}})^{-1}\mathbf{X}$, for which $\tilde{\mathbf{A}}_{\mathrm{rw}}\mathbf{X}$ is the first-order Taylor approximation, that is $\tilde{\mathbf{A}}_{\mathrm{rw}}\mathbf{X} \approx \bar{\mathbf{X}}$. By exchanging $\tilde{\mathbf{A}}_{\mathrm{rw}}$ with $\tilde{\mathbf{A}}_{\mathrm{sym}}$ we obtain the same form as the graph convolution, i.e., $\tilde{\mathbf{X}} = \tilde{\mathbf{A}}_{\mathrm{sym}}\mathbf{X} \approx \bar{\mathbf{X}}$. As such, graph convolution can be viewed as an approximate solution of (5), where it minimizes the variation over the graph structure while keeping the new representations close to the original.

The optimization problem in (5) facilitates a closer look to the oversmoothing problem of graph convolution. Ideally, we want to obtain smoothing over nodes within the same cluster, however avoid smoothing over nodes from different clusters. The objective in (5) dictates only the first goal via the graph-regularization term. It is thus prone to oversmoothing when convolutions are applied repeatedly. To circumvent the issue and fulfill both goals simultaneously, we can add a negative term such as the sum of distances between disconnected pairs as follows.

$$\min_{\bar{\mathbf{X}}} \sum_{i \in \mathcal{V}} \|\bar{\mathbf{x}}_i - \mathbf{x}_i\|_{\tilde{\mathbf{D}}}^2 + \sum_{(i,j) \in \mathcal{E}} \|\bar{\mathbf{x}}_i - \bar{\mathbf{x}}_j\|_2^2 - \lambda \sum_{(i,j) \notin \mathcal{E}} \|\bar{\mathbf{x}}_i - \bar{\mathbf{x}}_j\|_2^2 \tag{6}$$

where $\lambda$ is a balancing scalar to account for different volume and importance of the two goals.[2] By deriving the closed-form solution of (6) and approximating it with first-order Taylor expansion, one can get a revised graph convolution operator with hyperparameter $\lambda$. In this paper, we take a different route. Instead of a completely new graph convolution operator, we propose a general and efficient "patch", called PAIRNORM, that can be applied to any form of graph convolution having the potential of oversmoothing.

Let $\tilde{\mathbf{X}}$ (the output of graph convolution) and $\dot{\mathbf{X}}$ respectively be the input and output of PAIRNORM. Observing that the output of graph convolution $\tilde{\mathbf{X}} = \tilde{\mathbf{A}}_{\mathrm{sym}}\mathbf{X}$ only achieves the first goal, PAIRNORM serves as a normalization layer that works on $\tilde{\mathbf{X}}$ to achieve the second goal of keeping disconnected pair representations farther off. Specifically, PAIRNORM normalizes $\tilde{\mathbf{X}}$ such that the total pairwise squared distance $\mathrm{TPSD}(\dot{\mathbf{X}}) := \sum_{i,j \in [n]} \|\dot{\mathbf{x}}_i - \dot{\mathbf{x}}_j\|_2^2$ is the same as $\mathrm{TPSD}(\mathbf{X})$. That is,

$$\sum_{(i,j) \in \mathcal{E}} \|\dot{\mathbf{x}}_i - \dot{\mathbf{x}}_j\|_2^2 + \sum_{(i,j) \notin \mathcal{E}} \|\dot{\mathbf{x}}_i - \dot{\mathbf{x}}_j\|_2^2 = \sum_{(i,j) \in \mathcal{E}} \|\mathbf{x}_i - \mathbf{x}_j\|_2^2 + \sum_{(i,j) \notin \mathcal{E}} \|\mathbf{x}_i - \mathbf{x}_j\|_2^2 . \tag{7}$$

By keeping the total pairwise squared distance *unchanged*, the term $\sum_{(i,j) \notin \mathcal{E}} \|\dot{\mathbf{x}}_i - \dot{\mathbf{x}}_j\|_2^2$ is guaranteed to be at least as large as the original value $\sum_{(i,j) \notin \mathcal{E}} \|\mathbf{x}_i - \mathbf{x}_j\|_2^2$ since the other term $\sum_{(i,j) \in \mathcal{E}} \|\dot{\mathbf{x}}_i - \dot{\mathbf{x}}_j\|_2^2 \approx \sum_{(i,j) \in \mathcal{E}} \|\tilde{\mathbf{x}}_i - \tilde{\mathbf{x}}_j\|_2^2$ is shrunk through the graph convolution.

In practice, instead of always tracking the original value $\mathrm{TPSD}(\mathbf{X})$, we can maintain a *constant* TPSD value $C$ across all layers, where $C$ is a hyperparameter that could be tuned per dataset.

To normalize $\tilde{\mathbf{X}}$ to constant TPSD, we need to first compute $\mathrm{TPSD}(\tilde{\mathbf{X}})$. Directly computing TPSD involves $n^2$ pairwise distances that is $\mathcal{O}(n^2 d)$, which can be time consuming for large datasets.

---

[2]There exist other variants of (6) that achieve similar goals, and we leave the space for future exploration.

Equivalently, normalization can be done via a two-step approach where TPSD is rewritten as[3]

$$\text{TPSD}(\tilde{\mathbf{X}}) = \sum_{i,j \in [n]} \|\tilde{\mathbf{x}}_i - \tilde{\mathbf{x}}_j\|_2^2 = 2n^2 \left( \frac{1}{n} \sum_{i=1}^n \|\tilde{\mathbf{x}}_i\|_2^2 - \|\frac{1}{n} \sum_{i=1}^n \tilde{\mathbf{x}}_i\|_2^2 \right) . \tag{8}$$

The first term (ignoring the scale $2n^2$) in Eq. (8) represents the mean squared length of node representations, and the second term depicts the squared length of the mean of node representations. To simplify the computation of (8), we subtract the row-wise mean from each $\tilde{\mathbf{x}}_i$, i.e., $\tilde{\mathbf{x}}_i^c = \tilde{\mathbf{x}}_i - \frac{1}{n} \sum_i^n \tilde{\mathbf{x}}_i$ where $\tilde{\mathbf{x}}_i^c$ denotes the centered representation. Note that this shifting does *not* affect the TPSD, and furthermore drives the term $\|\frac{1}{n} \sum_{i=1}^n \tilde{\mathbf{x}}_i\|_2^2$ to zero, where computing TPSD$(\tilde{\mathbf{X}})$ boils down to calculating the squared Frobenius norm of $\tilde{\mathbf{X}}^c$ and overall takes $\mathcal{O}(nd)$. That is,

$$\text{TPSD}(\tilde{\mathbf{X}}) = \text{TPSD}(\tilde{\mathbf{X}}^c) = 2n \|\tilde{\mathbf{X}}^c\|_F^2 . \tag{9}$$

In summary, our proposed PAIRNORM (with input $\tilde{\mathbf{X}}$ and output $\dot{\mathbf{X}}$) can be written as a two-step, center-and-scale, normalization procedure:

$$\tilde{\mathbf{x}}_i^c = \tilde{\mathbf{x}}_i - \frac{1}{n} \sum_{i=1}^n \tilde{\mathbf{x}}_i \qquad \text{(Center)} \tag{10}$$

$$\dot{\mathbf{x}}_i = s \cdot \frac{\tilde{\mathbf{x}}_i^c}{\sqrt{\frac{1}{n} \sum_{i=1}^n \|\tilde{\mathbf{x}}_i^c\|_2^2}} = s\sqrt{n} \cdot \frac{\tilde{\mathbf{x}}_i^c}{\sqrt{\|\tilde{\mathbf{X}}^c\|_F^2}} \qquad \text{(Scale)} \tag{11}$$

After scaling the data remains centered, that is, $\|\sum_{i=1}^n \dot{\mathbf{x}}_i\|_2^2 = 0$. In Eq. (11), $s$ is a hyperparameter that determines $C$. Specifically,

$$\text{TPSD}(\dot{\mathbf{X}}) = 2n\|\dot{\mathbf{X}}\|_F^2 = 2n \sum_i \|s \cdot \frac{\tilde{\mathbf{x}}_i^c}{\sqrt{\frac{1}{n} \sum_i \|\tilde{\mathbf{x}}_i^c\|_2^2}}\|_2^2 = 2n \frac{s^2}{\frac{1}{n} \sum_i \|\tilde{\mathbf{x}}_i^c\|_2^2} \sum_i \|\tilde{\mathbf{x}}_i^c\|_2^2 = 2n^2 s^2 \tag{12}$$

Then, $\dot{\mathbf{X}} := \text{PAIRNORM}(\tilde{\mathbf{X}})$ has row-wise mean $\mathbf{0}$ (i.e., is centered) and constant total pairwise squared distance $C = 2n^2 s^2$. An illustration of PAIRNORM is given in Figure 2. The output of PAIRNORM is input to the next convolution layer.

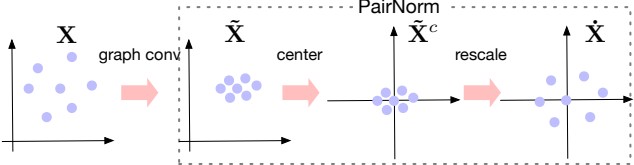

Figure 2: Illustration of PAIRNORM, comprising centering and rescaling steps.

We also derive a variant of PAIRNORM by replacing $\sum_{i=1}^n \|\tilde{\mathbf{x}}_i^c\|_2^2$ in Eq. (11) with $n\|\tilde{\mathbf{x}}_i^c\|_2^2$, such that the scaling step computes $\dot{\mathbf{x}}_i = s \cdot \frac{\tilde{\mathbf{x}}_i^c}{\|\tilde{\mathbf{x}}_i^c\|_2}$. We call it PAIRNORM-SI (for Scale Individually), which imposes more restriction on node representations, such that all have the same $L_2$-norm $s$. In practice we found that both PAIRNORM and PAIRNORM-SI work well for SGC, whereas PAIRNORM-SI provides better and more stable results for GCN and

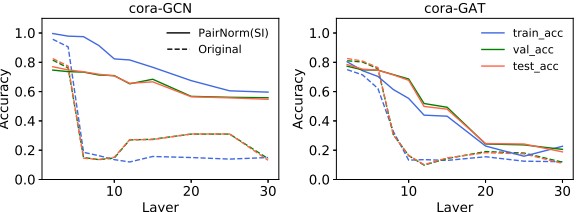

Figure 3: (best in color) Performance comparison of the original (dashed) vs. PAIRNORM-enhanced (solid) GCN and GAT models with increasing layers on `Cora`.

GAT. The reason why GCN and GAT require stricter normalization may be because they have more parameters and are more prone to overfitting. In Appx. A.6 we provide additional measures to demonstrate why PAIRNORM and PAIRNORM-SI work. In all experiments, we employ PAIRNORM for SGC and PAIRNORM-SI for both GCN and GAT.

PAIRNORM is effective and efficient in solving the oversmoothing problem of GNNs. As a general normalization layer, it can be used for any GNN. Solid lines in Figure 1 present the performance

---

[3]See Appendix A.1 for the detailed derivation.

of SGC on `Cora` with increasing number of layers, where we employ PAIRNORM after each graph convolution layer, as compared to 'vanilla' versions. Similarly, Figure 3 is for GCN and GAT (PAIRNORM is applied after the activation of each graph convolution). Note that the performance decay with PAIRNORM-at-work is much slower. (See Fig.s 5–6 in Appx. A.3 for other datasets.)

While PAIRNORM enables deeper models that are more robust to oversmoothing, it may seem odd that the overall test accuracy does not improve. In fact, the benchmark graph datasets often used in the literature require no more than 4 layers, after which performance decays (even if slowly). In the next section, we present a realistic use case setting for which deeper models are more likely to provide higher performance, where the benefit of PAIRNORM becomes apparent.

## 3.2 A CASE WHERE DEEPER GNNs ARE BENEFICIAL

In general, oversmoothing gets increasingly more severe as the number of layers goes up. A task would benefit from employing PAIRNORM more if it required a large number of layers to achieve its best performance. To this effect we study the "missing feature setting", where a subset of the nodes *lack* feature vectors. Let $\mathcal{M} \subseteq \mathcal{V}_u$ be the set where $\forall m \in \mathcal{M}, \mathbf{x}_m = \emptyset$, i.e., all of their features are missing. We denote with $p = |\mathcal{M}|/|\mathcal{V}_u|$ the missing fraction. We call this variant of the task as semi-supervised node classification with missing vectors (SSNC-MV). Intuitively, one would require a larger number of propagation steps (hence, a deeper GNN) to be able to "recover" effective feature representations for these nodes.

SSNC-MV is a general and realistic problem that finds several applications in the real world. For example, the credit lending problem of identifying low- vs. high-risk customers (nodes) can be modeled as SSNC-MV where a large fraction of nodes do not exhibit any meaningful features (e.g., due to low-volume activity). In fact, many graph-based classification tasks with the cold-start issue (entity with no history) can be cast into SSNC-MV. To our knowledge, this is the *first* work to study the SSNC-MV problem using GNN models.

Figure 4 presents the performance of SGC, GCN, and GAT models on `Cora` with increasing number of layers, where we remove feature vectors from all the unlabeled nodes, i.e. $p = 1$. The models with PAIRNORM achieve a higher test accuracy compared to those without, which they typically reach at a larger number of layers. (See Fig. 7 in Appx. A.4 for results on other datasets.)

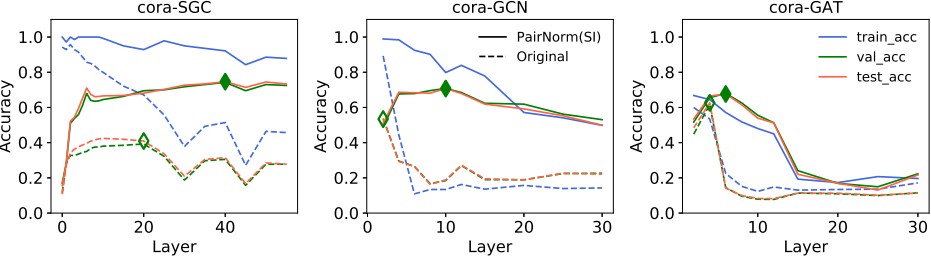

Figure 4: (best in color) Comparison of 'vanilla' vs. PAIRNORM-enhanced SGC, GCN, and GAT performance on `Cora` for $p = 1$. Green diamond symbols depict the layer at which validation accuracy peaks. PAIRNORM boosts overall performance by enabling more robust deep GNNs.

## 4 EXPERIMENTS

In section 3 we have shown the robustness of PAIRNORM-enhanced models against increasing number of layers in SSNC problem. In this section we design extensive experiments to evaluate the effectiveness of PAIRNORM under the SSNC-MV setting, over SGC, GCN and GAT models.

### 4.1 EXPERIMENT SETUP

**Datasets.** We use 4 well-known benchmark datasets in GNN domain: `Cora`, `Citeseer`, `Pubmed` (Sen et al., 2008), and `CoauthorCS` (Shchur et al., 2018). Their statistics are reported in Appx. A.2. For `Cora`, `Citeseer` and `Pubmed`, we use the same dataset splits as Kipf & Welling (2017), where all nodes outside train and validation are used as test set. For `CoauthorCS`, we randomly split all nodes into train/val/test as $3\%/10\%/87\%$, and keep the same split for all experiments.
**Models.** We use three different GNN models as our base model: SGC (Wu et al., 2019), GCN (Kipf & Welling, 2017), and GAT (Velickovic et al., 2018). We compare our PAIRNORM with residual connection method (He et al., 2016) over base models (except SGC since there is no "resid-

ual connected" SGC), as we surprisingly find it can slow down oversmoothing and benefit SSNC-MV problem. Similar to us, residual connection is a general technique that can be applied to any model without changing its architecture. We focus on the comparison between the base models and PAIRNORM-enhanced models, rather than achieving the state of the art performance for SSNC and SSNC-MV. There exist a few other work addressing oversmoothing (Klicpera et al., 2019; Li et al., 2018; Rong et al., 2019; Xu et al., 2018) however they design specialized architectures and not simple "patch" procedures like PAIRNORM that can be applied on top of any GNN.

**Hyperparameters.** We choose the hyperparameter $s$ of PAIRNORM from $\{0.1, 1, 10, 50, 100\}$ over validation set for SGC, while keeping it fixed at $s = 1$ for both GCN and GAT due to resource limitations. We set the #hidden units of GCN and GAT (#attention heads is set to 1) to 32 and 64 respectively for all datasets. Dropout with rate 0.6 and $L_2$ regularization with penalty $5 \cdot 10^{-4}$ are applied to GCN and GAT. For SGC, we vary number of layers in $\{1, 2, \ldots 10, 15, \ldots, 60\}$ and for GCN and GAT in $\{2, 4, \ldots, 12, 15, 20, \ldots, 30\}$.

**Configurations.** For PAIRNORM-enhanced models, we apply PAIRNORM after each graph convolution layer (i.e., after activation if any) in the base model. For residual-connected models with $t$ skip steps, we connect the output of $l$-th layer to $(l + t)$-th, that is, $\mathbf{H}_{\text{new}}^{(l+t)} = \mathbf{H}^{(l+t)} + \mathbf{H}^{(l)}$ where $\mathbf{H}^{(l)}$ denotes the output of $l$-th graph convolution (after activation). For the SSNC-MV setting, we randomly erase $p$ fraction of the feature vectors from nodes in validation and test sets (for which we input vector $\mathbf{0} \in \mathbb{R}^d$), whereas all training (labeled) nodes keep their original features (See 3.2). We run each experiment within 1000 epochs 5 times and report the average performance. We mainly use a single GTX-1080ti GPU, with some SGC experiments ran on an Intel i7-8700k CPU.

## 4.2 EXPERIMENT RESULTS

We first show the global performance gain of applying PAIRNORM to SGC for SSNC-MV under varying feature missing rates as shown in Table 1. PAIRNORM-enhanced SGC performs similar or better over 0% missing, while it significantly outperforms vanilla SGC for most other settings, especially for larger missing rates. #L denotes the best number of layers for the model that yields the largest average validation accuracy (over 5 runs), for which we report the average test accuracy (Acc). Notice the larger #L values for SGC-PN compared to vanilla SGC, which shows the power of PAIRNORM for enabling "deep" SGC models by effectively tackling oversmoothing.

Similar to Wu et al. (2019) who showed that the simple SGC model achieves comparable or better performance as other GNNs for various tasks, we found PAIRNORM-enhanced SGC to follow the same trend when compared with PAIRNORM-enhanced GCN and GAT, for all SSNC-MV settings. Due to its simplicity and extreme efficiency, we believe PAIRNORM-enhanced SGC sets a strong baseline for the SSNC-MV problem.

Table 1: Comparison of 'vanilla' vs. PAIRNORM-enhanced SGC performance in `Cora`, `Citeseer`, `Pubmed`, and `CoauthorCS` for SSNC-MV problem, with missing rate ranging from 0% to 100%. Showing test accuracy at $\#L$ ($K$ in Eq. 4) layers, at which model achieves best validation accuracy.

| Missing Percentage | | 0% | | 20% | | 40% | | 60% | | 80% | | 100% | |
|---|---|---|---|---|---|---|---|---|---|---|---|---|---|
| Dataset | Method | Acc | #L | Acc | #L | Acc | #L | Acc | #L | Acc | #L | Acc | #L |
| Cora | SGC | **0.815** | 4 | **0.806** | 5 | 0.786 | 3 | 0.742 | 4 | 0.733 | 3 | 0.423 | 15 |
| | SGC-PN | 0.811 | 7 | 0.799 | 7 | **0.797** | 7 | **0.783** | 20 | **0.780** | 25 | **0.745** | 40 |
| Citeseer | SGC | 0.689 | 10 | 0.684 | 6 | **0.668** | 8 | **0.657** | 9 | 0.565 | 8 | 0.290 | 2 |
| | SGC-PN | **0.706** | 3 | **0.695** | 3 | 0.653 | 4 | 0.641 | 5 | **0.590** | 50 | **0.486** | 50 |
| Pubmed | SGC | 0.754 | 1 | 0.748 | 1 | 0.723 | 4 | 0.746 | 2 | 0.659 | 3 | 0.399 | 35 |
| | SGC-PN | **0.782** | 9 | **0.781** | 7 | **0.778** | 60 | **0.782** | 7 | **0.772** | 60 | **0.719** | 40 |
| CoauthorCS | SGC | 0.914 | 1 | 0.898 | 2 | 0.877 | 2 | 0.824 | 2 | 0.751 | 4 | 0.318 | 2 |
| | SGC-PN | **0.915** | 2 | **0.909** | 2 | **0.899** | 3 | **0.891** | 4 | **0.880** | 8 | **0.860** | 20 |

We next employ PAIRNORM-SI for GCN and GAT under the same setting, comparing it with the residual (skip) connections technique. Results are shown in Table 2 and Table 3 respectively for GCN and GAT. Due to space and resource limitations, we only show results for 0% and 100% missing rate scenarios. (We provide results for other missing rates (70, 80, 90%) over 1 run only in Appx. A.5.) We observe similar trend for GCN and GAT: (1) vanilla model suffers from performance drop under SSNC-MV with increasing missing rate; (2) both residual connections and PAIRNORM-SI enable deeper models and improve performance (note the larger #L and Acc); (3) GCN-PN and

GAT-PN achieve performance that is comparable or better than just using skips; (4) performance can be further improved (albeit slightly) by using skips along with PAIRNORM-SI.[4]

Table 2: Comparison of 'vanilla' and (PAIRNORM-SI/ residual)-enhanced GCN performance on `Cora`, `Citeseer`, `Pubmed`, and `CoauthorCS` for SSNC-MV problem, with 0% and 100% feature missing rate. $t$ represents the skip-step of residual connection. (See A.5 Fig. 8 for more settings.)

| Dataset | Cora | | Citeseer | | Pubmed | | CoauthorCS | |
|---|---|---|---|---|---|---|---|---|
| Missing(%) | 0% | 100% | 0% | 100% | 0% | 100% | 0% | 100% |
| Method | Acc #L | Acc #L | Acc #L | Acc #L | Acc #L | Acc #L | Acc #L | Acc #L |
| GCN | 0.821 2 | 0.582 2 | 0.695 2 | 0.313 2 | 0.779 2 | 0.449 2 | 0.877 2 | 0.452 4 |
| GCN-PN | 0.790 2 | 0.731 10 | 0.660 2 | 0.498 8 | 0.780 30 | **0.745** 25 | 0.910 2 | **0.846** 12 |
| GCN-t1 | 0.822 2 | 0.721 15 | 0.696 2 | 0.441 12 | 0.780 2 | 0.656 25 | 0.898 2 | 0.727 12 |
| GCN-t1-PN | 0.780 2 | 0.724 30 | 0.648 2 | 0.465 10 | 0.756 15 | 0.690 12 | 0.898 2 | 0.830 20 |
| GCN-t2 | 0.820 2 | 0.722 10 | 0.691 2 | 0.432 20 | 0.779 2 | 0.645 20 | 0.882 4 | 0.630 20 |
| GCN-t2-PN | 0.785 4 | **0.740** 30 | 0.650 2 | **0.508** 12 | 0.770 15 | 0.725 30 | 0.911 2 | 0.839 20 |

Table 3: Comparison of 'vanilla' and (PAIRNORM-SI/ residual)-enhanced GAT performance on `Cora`, `Citeseer`, `Pubmed`, and `CoauthorCS` for SSNC-MV problem, with 0% and 100% feature missing rate. $t$ represents the skip-step of residual connection. (See A.5 Fig. 9 for more settings.)

| Dataset | Cora | | Citeseer | | Pubmed | | CoauthorCS | |
|---|---|---|---|---|---|---|---|---|
| Missing(%) | 0% | 100% | 0% | 100% | 0% | 100% | 0% | 100% |
| Method | Acc #L | Acc #L | Acc #L | Acc #L | Acc #L | Acc #L | Acc #L | Acc #L |
| GAT | 0.823 2 | 0.653 4 | 0.693 2 | 0.428 4 | 0.774 6 | 0.631 4 | 0.892 4 | 0.737 4 |
| GAT-PN | 0.787 2 | 0.718 6 | 0.670 2 | 0.483 4 | 0.774 12 | **0.714** 10 | 0.916 2 | 0.843 8 |
| GAT-t1 | 0.822 2 | 0.706 8 | 0.693 2 | 0.461 6 | 0.769 4 | 0.698 8 | 0.899 4 | 0.842 10 |
| GAT-t1-PN | 0.787 2 | 0.710 10 | 0.658 6 | 0.500 10 | 0.757 4 | 0.684 12 | 0.911 2 | 0.844 20 |
| GAT-t2 | 0.820 2 | 0.691 8 | s0.692 2 | 0.461 6 | 0.774 8 | 0.702 8 | 0.895 4 | 0.803 6 |
| GAT-t2-PN | 0.788 4 | **0.738** 12 | 0.672 4 | **0.517** 10 | 0.776 15 | 0.704 12 | 0.917 2 | **0.855** 30 |

## 5 RELATED WORK

**Oversmoothing in GNNs:** Li et al. (2018) was the first to call attention to the oversmoothing problem. Xu et al. (2018) introduced Jumping Knowledge Networks, which employ skip connections for multi-hop message passing and also enable different neighborhood ranges. Klicpera et al. (2019) proposed a propagation scheme based on personalized Pagerank that ensures locality (via teleports) which in turn prevents oversmoothing. Li et al. (2019) built on ideas from ResNet to use residual as well as dense connections to train deep GCNs. DropEdge Rong et al. (2019) proposed to alleviate oversmoothing through message passing reduction via removing a certain fraction of edges at random from the input graph. These are all specialized solutions that introduce additional parameters and/or a different network architecture.

**Normalization Schemes for Deep-NNs:** There exist various normalization schemes proposed for deep neural networks, including batch normalization Ioffe & Szegedy (2015), weight normalization Salimans & Kingma (2016), layer normalization Ba et al. (2016), and so on. Conceptually these have substantially different goals (e.g., reducing training time), and were not proposed for *graph* neural networks nor the oversmoothing problem therein. Important difference to note is that larger depth in regular neural-nets does not translate to more hops of propagation on a graph structure.

## 6 CONCLUSION

We investigated the oversmoothing problem in GNNs and proposed PAIRNORM, a novel normalization layer that boosts the robustness of deep GNNs against oversmoothing. PAIRNORM is fast to compute, requires no change in network architecture nor any extra parameters, and can be applied to any GNN. Experiments on real-world classification tasks showed the effectiveness of PAIRNORM, where it provides performance gains when the task benefits from more layers. Future work will explore other use cases of deeper GNNs that could further showcase PAIRNORM's advantages.

---

[4] Notice a slight performance drop when PAIRNORM is applied at 0% rate. For this setting, and the datasets we have, shallow networks are sufficient and smoothing through only a few (2-4) layers improves generalization ability for the SSNC problem (recall Figure 1 solid lines). PAIRNORM has a small reversing effect in these scenarios, hence the small performance drop.

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

# A  APPENDIX

## A.1  DERIVATION OF EQ. 8

$$\text{TPSD}(\tilde{\mathbf{X}}) = \sum_{i,j\in[n]} \|\tilde{\mathbf{x}}_i - \tilde{\mathbf{x}}_j\|_2^2 = \sum_{i,j\in[n]} (\tilde{\mathbf{x}}_i - \tilde{\mathbf{x}}_j)^T(\tilde{\mathbf{x}}_i - \tilde{\mathbf{x}}_j) \tag{13}$$

$$= \sum_{i,j\in[n]} (\tilde{\mathbf{x}}_i^T\tilde{\mathbf{x}}_i + \tilde{\mathbf{x}}_j^T\tilde{\mathbf{x}}_j - 2\tilde{\mathbf{x}}_i^T\tilde{\mathbf{x}}_j) \tag{14}$$

$$= 2n \sum_{i\in[n]} \tilde{\mathbf{x}}_i^T\tilde{\mathbf{x}}_i - 2 \sum_{i,j\in[n]} \tilde{\mathbf{x}}_i^T\tilde{\mathbf{x}}_j \tag{15}$$

$$= 2n \sum_{i\in[n]} \|\tilde{\mathbf{x}}_i\|_2^2 - 2\mathbf{1}^T\tilde{\mathbf{X}}\tilde{\mathbf{X}}^T\mathbf{1} \tag{16}$$

$$= 2n \sum_{i\in[n]} \|\tilde{\mathbf{x}}_i\|_2^2 - 2\|\mathbf{1}^T\tilde{\mathbf{X}}\|_2^2 \tag{17}$$

$$= 2n^2 \left( \frac{1}{n}\sum_{i=1}^{n} \|\tilde{\mathbf{x}}_i\|_2^2 - \|\frac{1}{n}\sum_{i=1}^{n} \tilde{\mathbf{x}}_i\|_2^2 \right) \ . \tag{18}$$

## A.2  DATASET STATISTICS

Table 4: Dataset statistics.

| Name | #Nodes | #Edges | #Features | #Classes | Label Rate |
|------|--------|--------|-----------|----------|------------|
| Cora | 2708 | 5429 | 1433 | 7 | 0.052 |
| Citeseer | 3327 | 4732 | 3703 | 6 | 0.036 |
| Pubmed | 19717 | 44338 | 500 | 3 | 0.003 |
| CoauthorCS | 18333 | 81894 | 6805 | 15 | 0.030 |

## A.3  ADDITIONAL PERFORMANCE PLOTS WITH INCREASING NUMBER OF LAYERS

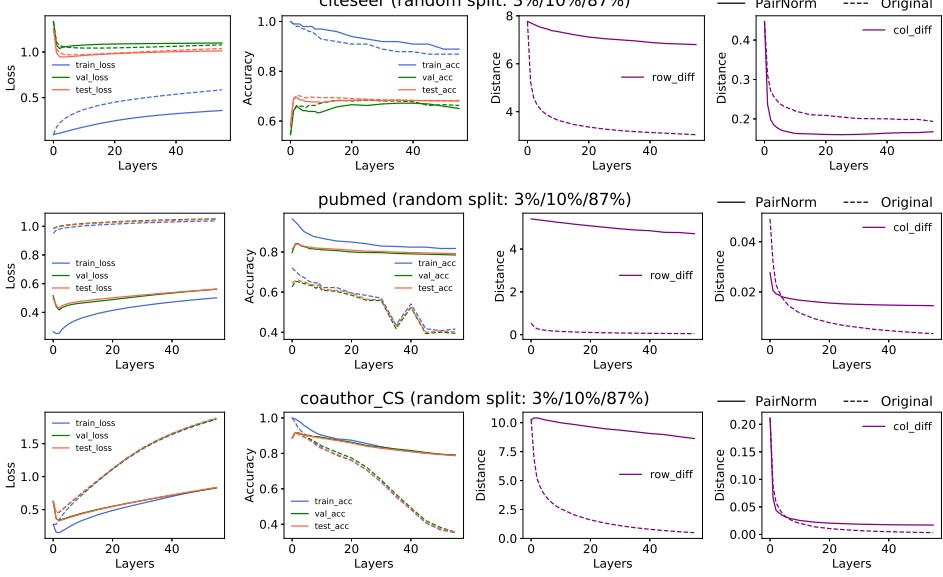

Figure 5: Comparison of 'vanilla' vs. PAIRNORM-enhanced SGC, corresponding to Figure 1, for datasets (from top to bottom) Citeseer, Pubmed, and CoauthorCS. PAIRNORM provides improved robustness to performance decay due to oversmoothing with increasing number of layers.

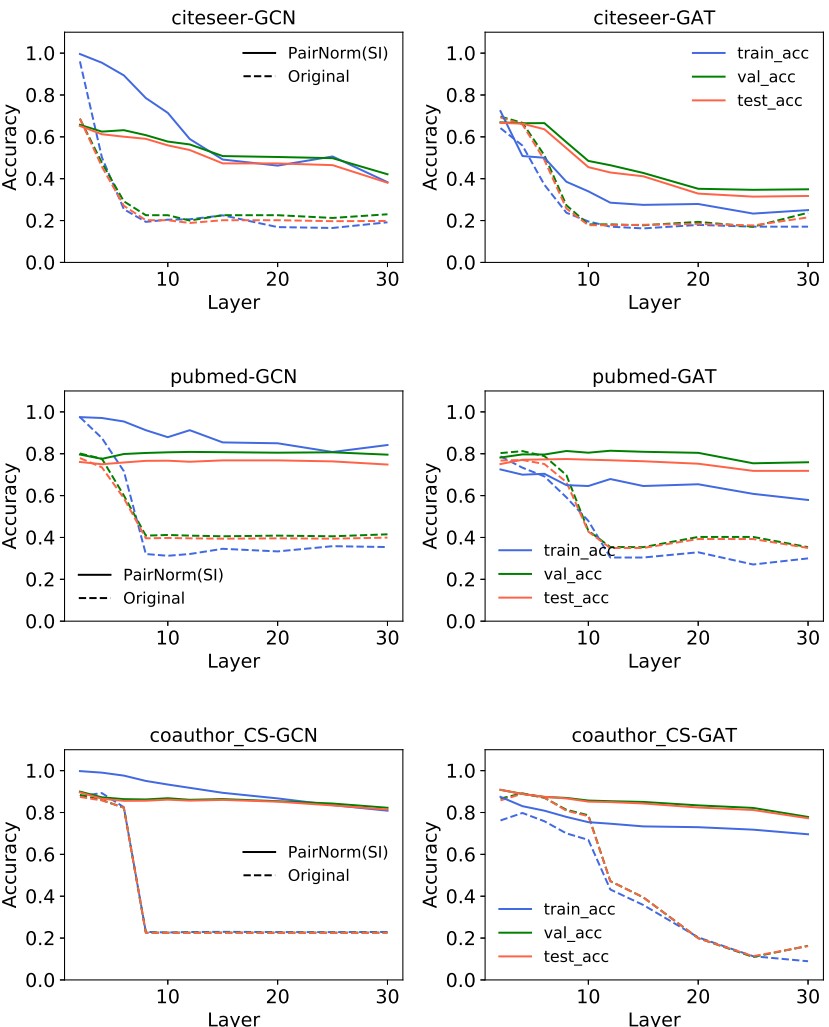

Figure 6: Comparison of 'vanilla' (dashed) vs. PAIRNORM-enhanced (solid) GCN (left) and GAT (right) models, corresponding to Figure 3, for datasets (from top to bottom) `Citeseer`, `Pubmed`, and `CoauthorCS`. PAIRNORM provides improved robustness against performance decay with increasing number of layers.

## A.4   ADDITIONAL PERFORMANCE PLOTS WITH INCREASING NUMBER OF LAYERS UNDER SSNC-MV WITH $p = 1$

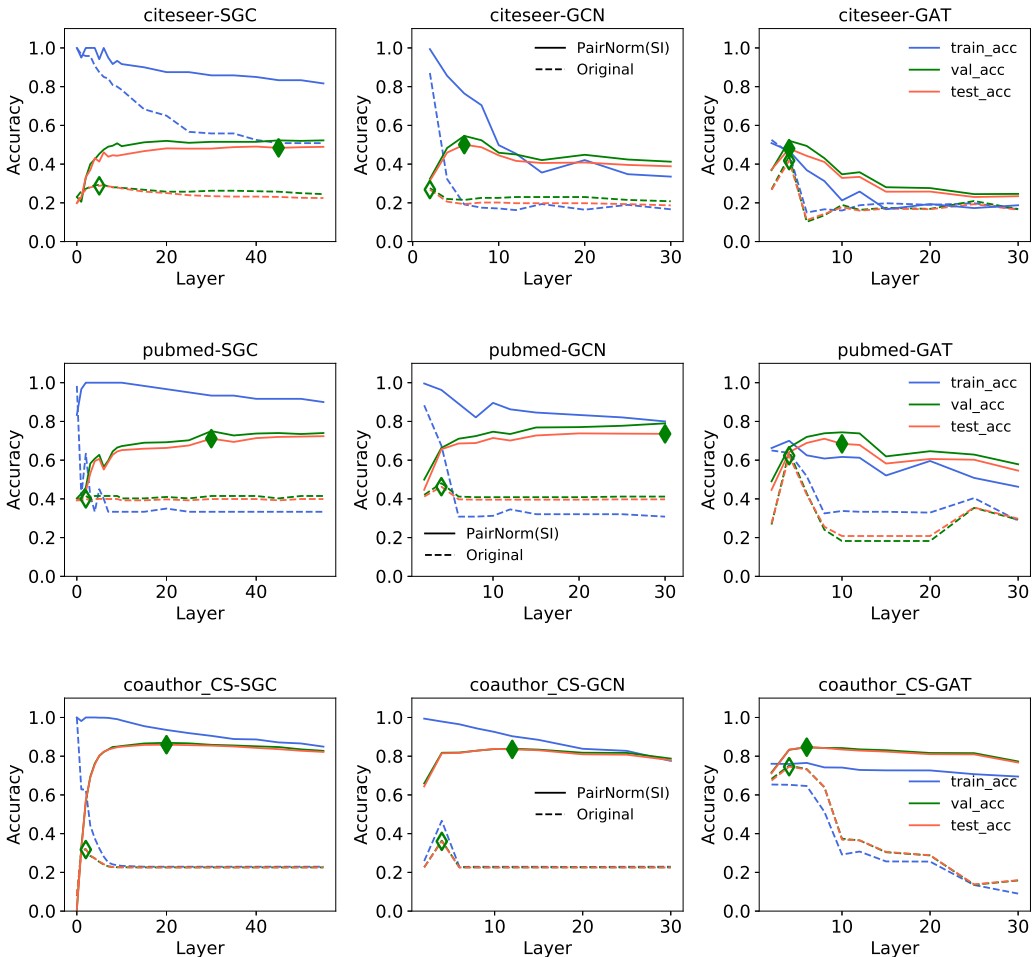

Figure 7:   Comparison of 'vanilla' (dashed) vs. PAIRNORM-enhanced (solid) (from left to right) SGC, GCN, and GAT model performance under SSNC-MV for $p = 1$, corresponding to Figure 4, for datasets (from top to bottom) Citeseer, Pubmed, and CoauthorCS. Green diamond symbols depict the layer at which validation accuracy peaks. PAIRNORM boosts overall performance by enabling more robust deep GNNs.

In this section we report additional experiment results under the SSNC-MV setting with varying missing fraction, in particular $p = \{0.7, 0.8, 0.9, 1\}$ and also report the base case where $p = 0$ for comparison.

Figure 8 presents results on all four datasets for GCN vs. PAIRNORM-enhanced GCN (denoted PN for short). The models without any skip connections are denoted by *-0, with one-hop skip connection by *-1, and with one and two-hop skip connections by *-2. Barcharts on the right report the best layer that each model produced the highest validation accuracy, and those on the left report the corresponding test accuracy. Figure 9 presents corresponding results for GAT.

We discuss the take-aways from these figures on the following page.

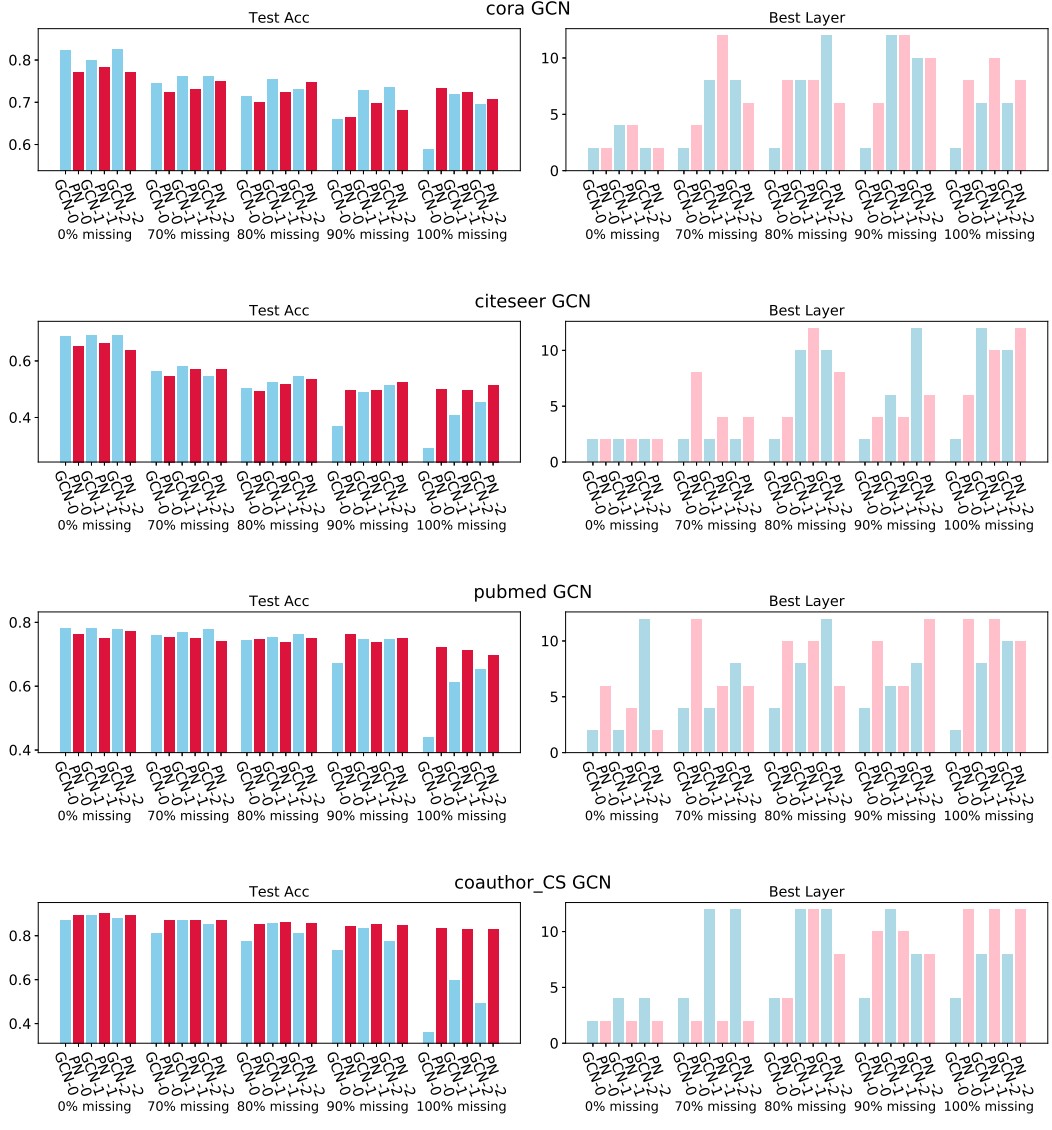

Figure 8: Supplementary results to Table 2 for GCN on (from top to bottom) `Cora`, `Citeseer`, `Pubmed`, and `CoauthorCS`.

We make the following observations based on Figures 8 and 9:

- Performance of 'vanilla' GCN and GAT models without skip connections (i.e., GCN-0 and GAT-0) drop monotonically as we increase missing fraction $p$.
- PAIRNORM-enhanced 'vanilla' models (PN-0, no skips) perform comparably or better than GCN-0 and GAT-0 in all cases, especially as $p$ increases. In other words, with PAIRNORM at work, model performance is more robust against missing data.
- Best number of layers for GCN-0 as we increase $p$ only changes between 2-4. For GAT-0, it changes mostly between 2-6.
- PAIRNORM-enhanced 'vanilla' models (PN-0, no skips) can go deeper, i.e., they can leverage a larger range of #layers (2-12) as we increase $p$. Specifically, GCN-PN-0 (GAT-PN-0) uses equal number or more layers than GCN-0 (GAT-0) in almost all cases.
- Without any normalization, adding skip connections helps—GCN/GAT-1 and GCN/GAT-2 are better than GCN/GAT-0, especially as we increase $p$.
- With PAIRNORM but no-skip, performance is comparable or better than just adding skips.
- Adding skips on top of PAIRNORM does not seem to introduce any notable gains.

In summary, simply employing our PAIRNORM for GCN and GAT provides robustness against oversmoothing that allows them to go deeper and achieve improved performance under SSNC-MV.

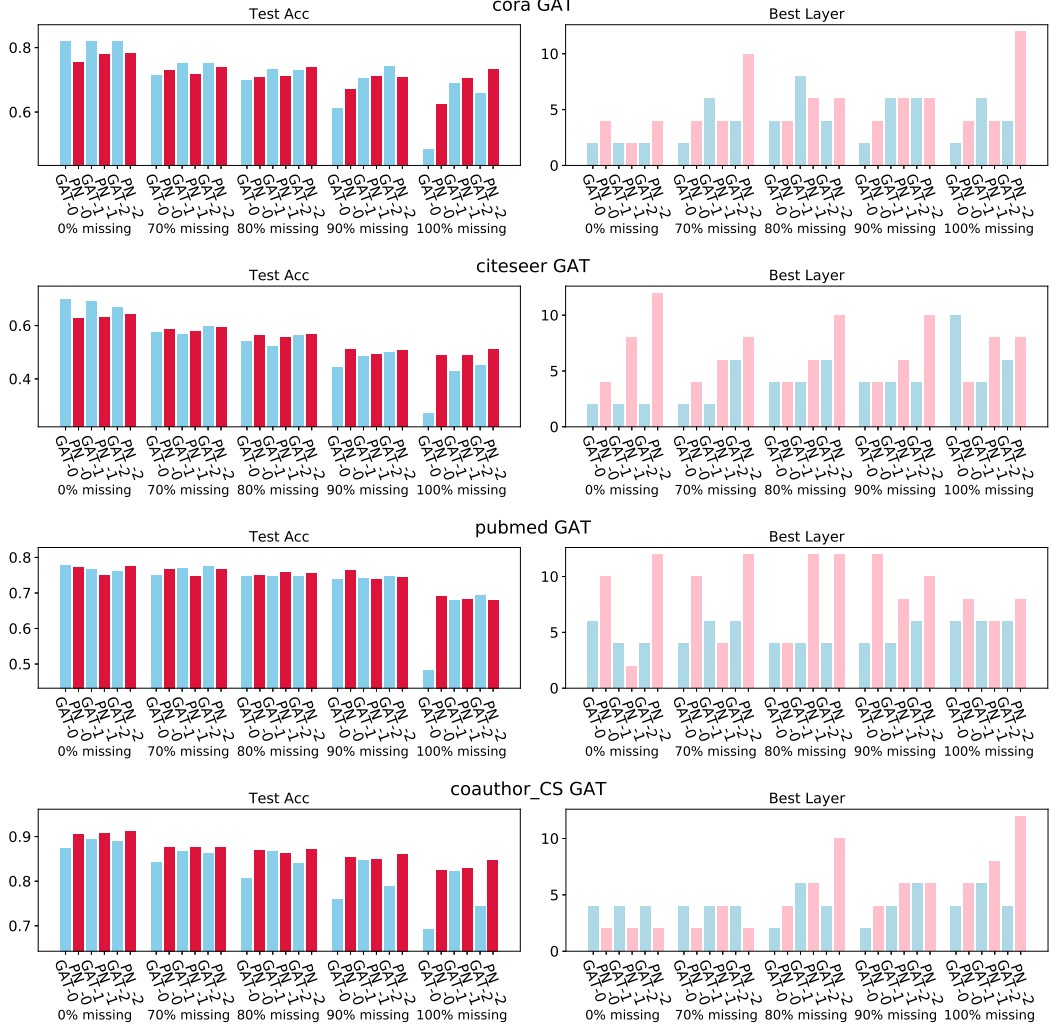

Figure 9: Supplementary results to Table 3 for GAT on (from top to bottom) `Cora`, `Citeseer`, `Pubmed`, and `CoauthorCS`.

### A.6   CASE STUDY: ADDITIONAL MEASURES FOR PAIRNORM AND PAIRNORM-SI WITH SGC AND GCN

To better understand why PAIRNORM and PAIRNORM-SI are helpful for training deep GNNs, we report additional measures for (SGC and GCN) with (PAIRNORM and PAIRNORM-SI) over the `Cora` dataset. In the main text, we claim TPSD (total pairwise squared distances) is constant across layers for SGC with PAIRNORM (for GCN/GAT this is not guaranteed because of the influence of activation function and dropout layer). In this section we empirically measure pairwise (squared) distances for both SGC and GCN, with PAIRNORM and PAIRNORM-SI.

#### A.6.1   SGC WITH PAIRNORM AND PAIRNORM-SI

To verify our analysis of PAIRNORM for SGC, and understand how the variant of PAIRNORM (PAIRNORM-SI) works, we measure the average pairwise squared distance (APSD) as well as the average pairwise distance (APD) between the representations for two categories of node pairs: (1) connected pairs (nodes that are directly connected in graph) and (2) random pairs (uniformly randomly chosen among the node set). APSD of random pairs reflects the TPSD, and APD of random pairs reflects the total pairwise distance (TPD). Under the homophily assumption of the labels w.r.t. the graph structure, we want APD or APSD of connected pairs to be small while keeping APD or APSD of random pairs relatively large.

The results are shown in Figure 10. Without normalization, SGC suffers from fast diminishing APD and APSD of random pairs. As we have proved, PAIRNORM normalizes APSD to be constant across layers, however it does not normalize APD, which appears to decrease linearly with increasing number of layers. Surprisingly, although PAIRNORM-SI is not theoretically proved to have a constant APSD and APD, empirically it achieves more stable APSD and APD than PAIRNORM. We were not able to prove this phenomenon mathematically, and leave it for further investigation.

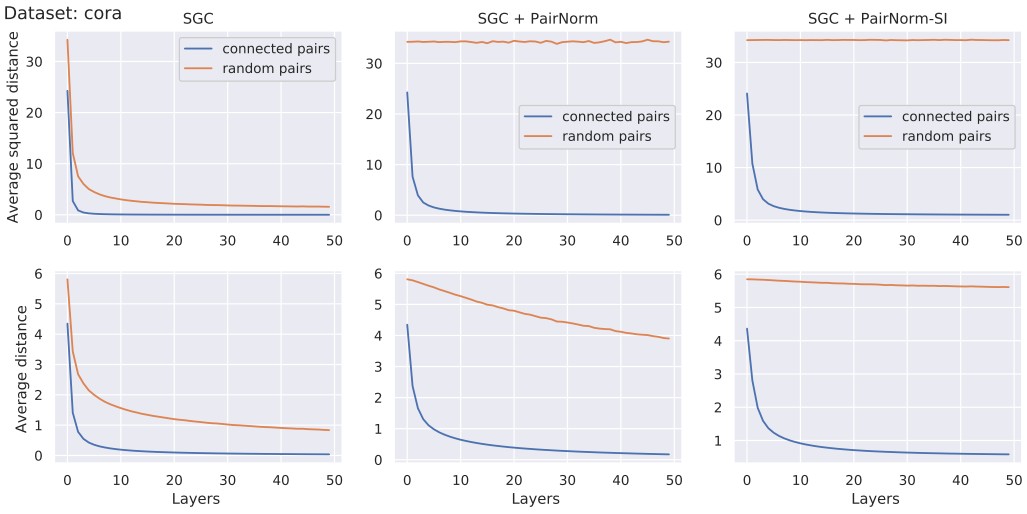

Figure 10: Measuring average distance (squared and not-squared) between representations at each layer for SGC, SGC with PAIRNORM, and SGC with PAIRNORM-SI. The setting is the same with Figure 1 and they share the same performance.

APD does not capture the full information of the distribution of pairwise distances. To show how the distribution changes by increasing number of layers, we use Tensorboard to plot the histograms of pairwise distances, as shown in Figure 11. Comparing SGC and SGC with PAIRNORM, adding PAIRNORM keeps the left shift (shrinkage) of the distribution of random pair distances much slower than without normalization, while still sharing similar behavior of the distribution of connected pairwise distances. PAIRNORM-SI seems to be more powerful in keeping the median and mean of the distribution of random pair distances stable, while "spreading" the distribution out by increasing the variance. The performance of PAIRNORM and PAIRNORM-SI are similar, however it seems that PAIRNORM-SI is more powerful in stabilizing TPD and TPSD.

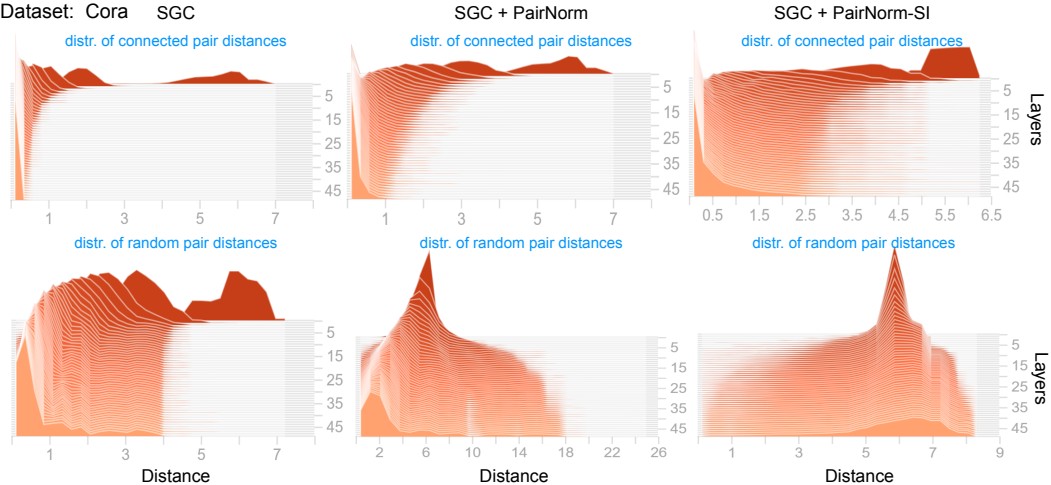

Figure 11: Measuring distribution of distances between representations at each layer for SGC, SGC with PAIRNORM, and SGC with PAIRNORM-SI. Supplementary results for Figure 10.

### A.6.2   GCN WITH PAIRNORM AND PAIRNORM-SI

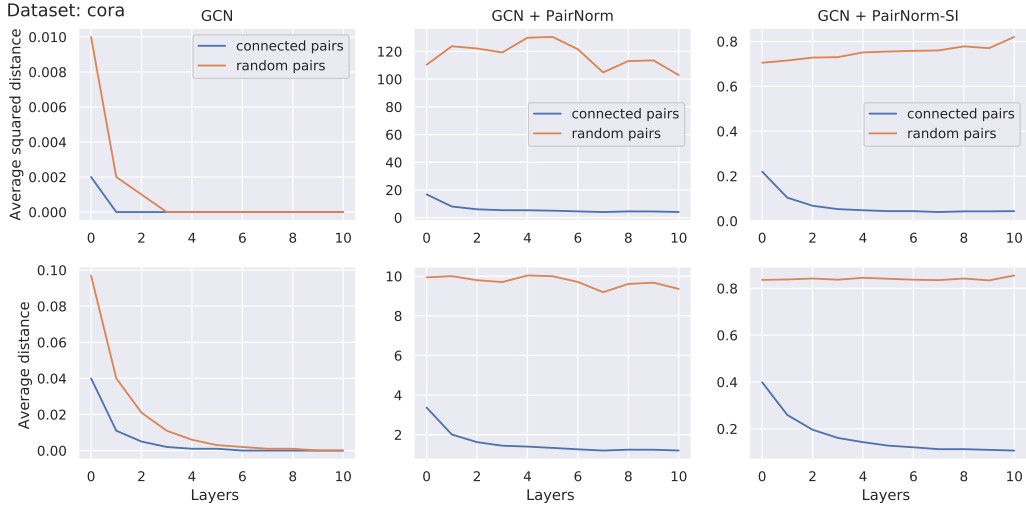

Figure 12: Measuring average distance (squared and not-squared) between representations at each layer for GCN, GCN with PAIRNORM, and GCN with PAIRNORM-SI. We trained three 12-layer GCNs with #hidden=128 and dropout=0.6 in 1000 epochs. Respective test set accuracies are 31.09%, 77.77%, 75.09%. Note that the scale of distances is not comparable across models, since they have learnable parameters that scale these distances differently.

The formal analysis for PAIRNORM and PAIRNORM-SI is based on SGC. GCN (and other GNNs) has learnable parameters, dropout layers, and activation layers, all of which complicate direct mathematical analyses. Here we perform similar empirical measurements for pairwise distances to get a rough sense of how PAIRNORM and PAIRNORM-SI work with GCN based on the `Cora` dataset. Figures 12 and 13 demonstrate how PAIRNORM and PAIRNORM-SI can help train a relatively deep (12 layers) GCN.

Notice that oversmoothing occurs very quickly for GCN without any normalization, where both connected and random pair distances reach zero (!). In contrast, GCN with PAIRNORM or PAIRNORM-SI is able to keep random pair distances relatively apart while allowing connected pair distances to shrink. As also stated in main text, using PAIRNORM-SI for GCN and GAT is relatively more

stable than using PAIRNORM in general cases (notice the near-constant random pair distances in the rightmost subfigures). There are several possible explanations for why PAIRNORM-SI is more stable. First, as shown in Figure 10 and Figure 12, PAIRNORM-SI not only keeps APSD stable but also APD, further, the plots of distributions of pairwise distances (Figures 11 and 13) also show the power of PAIRNORM-SI (notice the large gap between smaller connected pairwise distances and the larger random pairwise distances). Second, we conjecture that restricting representations to reside on a sphere can make training stable and faster, which we also observe empirically by studying the training curves. Third, GCN and GAT tend to overfit easily for the SSNC problem, due to many learnable parameters across layers and limited labeled input data, therefore it is possible that adding more restriction on these models helps reduce overfitting.

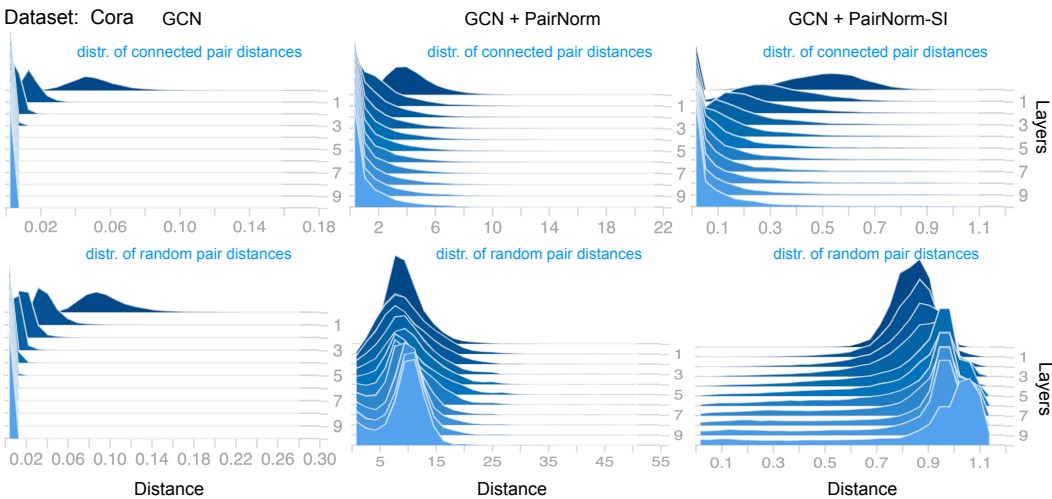

Figure 13: Measuring distribution of distances between representations at each layer for GCN, GCN with PAIRNORM, and GCN with PAIRNORM-SI. Supplementary results for Figure 12.

All in all, these empirical measurements as illustrated throughout the figures in this section demonstrates that PAIRNORM and PAIRNORM-SI successfully address the oversmoothing problem for deep GNNs. Our work is the *first* to propose a normalization layer specifically designed for *graph* neural networks, which we hope will kick-start more work in this area toward training more robust and effective GNNs.

