# OpenReview forum: "PairNorm: Tackling Oversmoothing in GNNs"
_ICLR.cc/2020/Conference — Accept (Poster)_

### Official Review · AnonReviewer2 · 2019-10-23
**Official Blind Review #2**

**Rating:** 8

**Review:**

Summary

It is known that GNNs are vulnerable to the oversmoothing problem, in which feature vectors on nodes get closer as we increase the number of (message passing type graph convolution layers). This paper proposed PairNorm, which is a normalization layer for GNNs to tackle this problem. The idea is to pull apart feature vectors on a pair of non-adjacent nodes (based on the interpretation of Laplace-type smoothing by NT and Maehara (2019)). To achieve this approximately with low computational complexity, PairNorm keeps the sum of distances of feature vectors on all node pairs approximately the same throughout layers. The paper conducted empirical studies to evaluate the effectiveness of the method. PairNorm improved the prediction performance and enabled make GNNs deep, especially when feature vectors are missing in the large portion of nodes (the SSNC-MV problem).


Decision

I want to recommend to accept the paper because, in my opinion, this paper contributes to deepening our understanding of graph NNs by giving new insights into what causes the oversmoothing problem and which types problem (deep) graph NNs can solve.
The common myth about graph NNs is that they cannot make themselves deep due to the oversmoothing. Therefore, oversmoothing is one of the big problems in the graph NN field and has been paid attention from both theoretical and empirical sides. This paper found that the deep structures do help to improve (or at least worsen) the predictive performance when the significant portion of nodes in a graph does not have input signals. To the best of our knowledge, this is the first paper that showed the effectiveness of deep structures in citation network datasets (Deep GCNs [Li et al., 2019] successfully improved the prediction performance of (residual) graph NNs using as many as 56 layers for point cloud datasets). The proposed method is theoretically backboned, easy to implement, and applicable to (theoretically) any graph NNs. Taking these things into account, I would like to judge the contribution of this paper is sufficiently significant to accept.


Minor Comments

	- Table 3. Remove s in the entry for GAT-t2 Citeseer 0%.


Questions

	- Can we interpret PairNorm (or the optimization problem (6)) from the viewpoint of graph spectra?
	- Although the motivation of Centering (10) is to ease the computation of TPD, I am curious how this operation contributes to performance. Since the constant signal does not have information for distinguishing nodes, eliminating it by Centering might result in emphasizing the signal component for nodes classification tasks. From a spectral point of view, Centering corresponds to eliminating the lowest frequency of a signal.
	- Figures 3 and 7 have shown that GCN and GAT did not perform well compared to SGC when the layer size increases. The authors discussed that this is because GCN and GAT are easier to overfit. However, SGC chose the hyperparameter $s$ from $\{0.1,1,10,50,100\}$, whereas the authors examined a single $s$ for GCN and GAT. Therefore, I think there is another hypothesis that simply the choice $s$ was misspecified. If this is the case, I am interested in the effect of $s$ on predictive performance.

[Li et al., 2018] Li, Qimai, Zhichao Han, and Xiao-Ming Wu. "Deeper insights into graph convolutional networks for semi-supervised learning." Thirty-Second AAAI Conference on Artificial Intelligence. 2018.

**Experience Assessment:**

I have published one or two papers in this area.

**Review Assessment: Checking Correctness Of Derivations And Theory:**

I carefully checked the derivations and theory.

**Review Assessment: Checking Correctness Of Experiments:**

I assessed the sensibility of the experiments.

**Review Assessment: Thoroughness In Paper Reading:**

I read the paper at least twice and used my best judgement in assessing the paper.

---

> ### Author Response · Authors · 2019-11-15
> **Thank you for liking our work**
>
> Thank you very much for reading our paper thoroughly and giving constructive feedback, and we are glad that you found our paper interesting and contributing to a deeper understanding of the field. We respond to your questions one-by-one in the following.
>
> >> Can we interpret PairNorm (or the optimization problem (6)) from the viewpoint of graph spectra?
>
> That is a great question that we have not thought about before. We are doing new work towards understanding stacking GraphConv operations in the spectral domain, but currently we do not have a complete answer for your question. To give some initial thought: The operation is working on features directly. Since it does not change the graph structure, it does not affect the eigenvectors or spectrum of the graph. However, it will affect the alignment/interaction between structure and features. Understanding the fusion between graph structure and features in spectral domain should be investigated more carefully.
>
>
> >> Although the motivation of Centering (10) is to ease the computation of TPD, I am curious how this operation contributes to performance.
>
> We have tested adding the mean back after Scale operation, and for SGC the performance remained the same. However for GCN and GAT, because of the activation function there will be a big difference. Empirically, they have similar performance but sometimes one is better and the other is worse. One does not seem to dominate the other.
>
>
> >>  Therefore, I think there is another hypothesis that simply the choice $s$ was misspecified. If this is the case, I am interested in the effect of $s$ on predictive performance.
>
>  We did several tests using different $s$ for the SSNC problem, and we found that $s$ does not affect performance much for GCN and GAT. We think this is because the parameter learning has some connection with the scale, so setting different $s$ is not that important. We do not have enough time for doing a thorough testing for all settings, as GCN and GAT are much slower to train than SGC. To sum up, we think it is not surprising that SGC works very well for these settings, which is also demonstrated in the original SGC paper (Wu et al., 2019).

---

### Official Review · AnonReviewer1 · 2019-10-29
**Official Blind Review #1**

**Rating:** 3

**Review:**

The article "PairNorm: Tackling Oversmoothing in GNNs" considers the interesting phenomenon of performance degradation of graph neural network when the depth of the network increases beyond the values of 2-4. The authors argue that one of the reasons for such behavior is so-called "oversmoothing", when intermediate representations become similar for all the nodes in the graph. The authors propose the special NN layer "PairNorm", which aims to battle with this issue.

The proposed PairNorm approach boils down to the recentering and normalization of all the representations after each graph-convolutional layer of the network. The authors consider 2 variants of choosing normalization constant:
1. The one which multiplies all the embeddings for the layer by the same number. This operation allows to keep the average squared pairwise distance between node representations constant.
2. The one which makes the norms of all the representations equal to pre-specified constant, i.e. just projection of all the embeddings on the sphere.

I should note that the two proposed approaches are very different in nature, though the latter one is introduced without much additional discussion. The benefits of approach 1 are not entirely clear as it basically just scales the whole embedding population. Such a scaling doesn't affect the relative distances between points and thus should not have major effect on the performance. Approach 2 is completely different due to the projection on the sphere of each embedding independently. However, the reasons why it is a good idea or not are not discussed in the paper.

The experimental part of the paper considers several standard graph data sets. The authors report that the proposed normalization schemes do not improve the quality of classification in the standard semi-supervised learning setting. They additionally consider artificially created missing features and observe increasing quality in such a scenario.

To sum up, I think that while the motivation behind the paper is very natural, it doesn't look like the paper finds the solution both theoretically and experimentally.

**Experience Assessment:**

I have read many papers in this area.

**Review Assessment: Checking Correctness Of Derivations And Theory:**

I carefully checked the derivations and theory.

**Review Assessment: Checking Correctness Of Experiments:**

I assessed the sensibility of the experiments.

**Review Assessment: Thoroughness In Paper Reading:**

I read the paper at least twice and used my best judgement in assessing the paper.

---

> ### Author Response · Authors · 2019-11-15
> **We have added new section in the paper to address your questions**
>
> Thank you for giving us feedback and raising questions that help us clarify our work further. We have done additional measurements and experiments to address your concerns, and these results are included in the last section of the Appendix (please see A.6 in the updated paper).
>
> We address the reviewer's questions one-by-one in the following.
>
> >> "The benefits of approach 1 are not entirely clear as it basically just scales the whole embedding population. Such a scaling doesn't affect the relative distances between points and thus should not have major effect on the performance."
>
> The second step (Eq. 11) of PairNorm scales the length of each node representation by multiplying by a scalar calculated from all node representations, the L2-distance between any two node representations is also scaled accordingly. If we understand correctly, the "relative distances among points" is referring to the ratio between any two distances. We claim that although PairNorm itself does not change the ratio, GraphConv + PairNorm does change it: the ratios are not the same across all layers’ representations. In Appendix A.6 we also empirically plot the distributions of pairwise distances, where one could see how all pairwise distances change with increasing number of layers.
>
> >> "Approach 2 is completely different due to the projection on the sphere of each embedding independently. However, the reasons why it is a good idea or not are not discussed in the paper."
>
> PairNorm-SI (approach 2) achieves a similar goal of making total pairwise distances stable, by adding more restrictions: instead of normalizing by the sum of all squared lengths, it normalizes any squared length directly. It is true that total pairwise squared distances is not exactly constant mathematically in that case, however approach 2 nevertheless does keep it stable empirically. We have put a lot effort to analyzing PairNorm-SI in the new section A.6 of the Appendix — please have a look and read it through; we believe it provides the the answer you are looking for.
>
> >> "The authors report that the proposed normalization schemes do not improve the quality of classification in the standard semi-supervised learning setting."
>
> Yes, this is correct. We find that solving oversmoothing problem would not improve the performance of SSL on the standard benchmark datasets. The reasons are two-fold. First, the best performance of SSL on standard datasets is achieved within less than 3 layers, at which oversmoothing does not happen. Rather, we obtain a smoothing effect, which is in fact beneficial. In such cases, clearly PairNorm is not needed as it is designed to solve the oversmoothing issue for deep GNNs. Second, smoothing is the key effect of Graph Convolution to achieve good performance for SSL, as it improves the generalization ability of the model by reducing the gap between training loss and validation loss. This is clearly shown in Figure 1, where the gap between training loss and validation/test loss shrinks with increasing number of layers. Generalization ability is the most important factor for improving performance, and this is true particularly for SSL where we only have a very small training set, which makes the empirical risk not reliable for estimating true risk. Empirically, we often see the training loss for SSL goes to 0 easily while validation loss is still large.
>
> >> "They additionally consider artificially created missing features and observe increasing quality in such a scenario."
>
>  We should state that although SSNC-MF is created by randomly removing features, this scenario is generally existing in the real-world. We have given example scenarios in our paper, another example would be privacy-related problems: for training ML algorithms, many companies can only release/use small fraction of users' data based on the privacy agreement. PairNorm is designed to solve oversmoothing, and SSNC-MF is such a problem where oversmoothing does happen as this scenarios necessitates training deep GNNs. While for SSL on standard datasets oversmoothing has no relationship with the best performance, in order to show the power and ability of PairNorm at solving oversmoothing, we needed to showcase a scenario where oversmoothing hurts the best possible performance.
>
> _____
> We hope our answers sufficiently addresses your concerns. To wrap up, we would like to to re-emphasize the contributions of PairNorm:
>
> 1. Solving oversmoothing problem and making training deep GNNs possible for the node-classification scenario,  having solid theoretical analysis over SGC.
> 2. PairNorm is a general "patch" -- applicable to any GNN. It can also be applied in any layer, even if say we change the graph structure at each layer.
> 3. PairNorm is the first normalization layer specifically designed for graph neural networks. We hope that more researchers can delve into this area.
> 4. We are also the first to investigate a new scenario, such as the SSNC-MF problem.

---

### Decision · Program_Chairs · 2019-12-19

**Decision:**

Accept (Poster)

**Comment:**

The paper proposes a way to tackle oversmoothing in Graph Neural Networks. The authors do a good job of motivating their approach, which is straightforward and works well. The paper is well written and the experiments are informative and well carried out. Therefore, I recommend acceptance. Please make suree thee final version reflects the discussion during the rebuttal.